# Prevalence of multidrug resistance *Salmonella* species isolated from clinical specimens at University of Gondar comprehensive specialized hospital Northwest Ethiopia: A retrospective study

**Azanaw Amare** [1]*, **Fekadu Asnakew**[2], **Yonas Asressie**[2], **Eshetie Guadie**[2], **Addisu Tirusew**[2], **Silenat Muluneh**[2], **Abebew Awoke**[2], **Muluneh Assefa**[1], **Worku Ferede**[2], **Alem Getaneh**[1], **Mulualem Lemma**[3]

1 Department of Medical Microbiology, School of Biomedical and Laboratory Sciences, College of Medicine and Health Sciences, University of Gondar, Gondar, Ethiopia, 2 School of Biomedical and Laboratory Sciences, College of Medicine and Health Sciences, University of Gondar, Gondar, Ethiopia, 3 Department of Immunology and Molecular Biology, School of Biomedical and Laboratory Sciences, College of Medicine and Health Sciences, University of Gondar, Gondar, Ethiopia

* azanaw03@gmail.com

## Abstract

### Background

Multidrug resistance Salmonellosis remains an important public health problem globally. The disease is among the leading causes of morbidity and mortality in developing countries, but there have been limited recent studies about the prevalence, antimicrobial resistance, and multidrug resistance patterns of *Salmonella* isolates from various clinical specimens.

### Objective

Aimed to assess the prevalence, antimicrobial resistance, and multidrug resistance patterns of *Salmonella* isolates from clinical specimens at the University of Gondar Comprehensive Specialised Hospital, northwestern Ethiopia.

### Method

A retrospective hospital-based cross-sectional study was conducted to determine the prevalence, antimicrobial resistance, and multidrug resistance patterns of isolated from all clinical specimens at the University of Gondar *Salmonella* Comprehensive Specialised Hospital from June 1st, 2017 to June 3rd, 2022. A total of 26,154 data points were collected using a checklist of records of laboratory registration. Clinical specimens were collected, inoculated, and incubated for about a week with visual inspection for growth and gram staining. The isolates were grown on MacConkey agar and Xylose Lysine Deoxycholate agar. Pure colonies were identified with a conventional biochemical test, and those unidentified at the species level were further identified by the analytical profile index-20E. Then, antimicrobial

**Data Availability Statement:** Authors to make all data necessary publicly available without restriction at the time of publication.

**Funding:** The author(s) received no specific funding for this work.

**Competing interests:** The authors have declared that no competing interests exist

**Abbreviations:** AMR, antimicrobial resistance; ATCC, American type culture collection; CLSI, Clinical and Laboratory Standard Institute; UoGCSH, University of Gondar Comprehensive Specialised Hospital; MDR, multidrug-resistant; MAC, MacConkey; XLD, Xylose Lysine Deoxycholate agar.

susceptibility was determined by the Kirby-Bauer disc diffusion technique. The multidrug resistance *Salmonella* isolates was identified using the criteria set by Magiorakos. Finally, the data was cleaned and checked for completeness and then entered into SPSS version 26 for analysis. Then the results were displayed using tables and figures.

## Results

Of the total 26,154 *Salmonella* suspected clinical samples, 41 (0.16%) *Salmonella* species were isolated. Most of the *Salmonella* isolates, 19 (46.3%), were in the age group of less than 18 years, followed by the age group of 19–44 years, 11 (26.8%). In this study, *S. enterica subsp. arizonae* accounts for the highest 21 (51%), followed by *S. paratyphi* A 9 (22%). Of the Salmonella isolates, *S. typhi* were highly resistant to ampicillin (100%), followed by tetracycline and trimethoprim-sulfamethoxazole, each accounting for 83.3%. Furthermore, *S. paratyphi* A was resistant to ampicillin (100%), tetracycline (88.9%), and chloramphenicol (88.9%). The overall multi-drug resistance prevalence was 22 (53.7%; 95% CI: 39.7–61). Accordingly, *S. paratyphi* A was 100% multidrug-resistant, followed by *S. typhi* (66.6%).

## Conclusion

A low prevalence of *Salmonella* species was observed in the past six years. Moreover, most *S. typhi* and *S. paratyphi* strains in the study area were found to be resistant to routinely recommended antibiotics like ciprofloxacin and ceftriaxone, compared to what was reported earlier. In addition, all isolates of *S. paratyphi* A and the majority of *S. typhi* were multidrug resistant. Therefore, health professionals should consider antimicrobial susceptibility tests and use antibiotics with caution for Salmonellosis management.

## Introduction

All around the world, especially in developing countries, salmonellosis is one of the leading causes of illness and death caused by *Salmonella* species [1–3]. The species can be classified as *Salmonella enterica* and *Salmonella bongori*, and these strains can be divided into subspecies based on biochemical and genomic studies, which include *enterica*, *salamae*, *arizonae*, *diarizonae*, *houtenae*, and *indica* [4,5].

*Salmonella* species are found in the intestines of animals, particularly birds, and poultry, and they spread from animals to humans through contaminated food [6]. These species invade the intestinal mucosa after getting into a person's stomach and colonizing the small and large intestines. Once the bacteria have proliferated, they can enter the lymphoid tissues of the gastrointestinal system and spread to the circulation. The bloodstream is where the *Salmonella* species can enter and spread to the liver, kidneys, and other organs [4]. The three main infectious diseases caused by *Salmonella* species infection in humans are typhoid fever, paratyphoid fever, and non-typhoidal *Salmonella*, which are all characterized by gastroenteritis and its complications, including septicemia, immunological signs, leucopenia, and neurological symptoms [7,8].

According to World Health Organization estimates, *Salmonella* is thought to be the cause of 3 million deaths globally each year [9]. *Salmonella* can cause typhoid fever (by *Salmonella Typhi* or *Paratyphi* A*)* and gastroenteritis or nontyphoidal salmonellosis (by *Salmonella*

*Typhimurium* or *Salmonella Enteritidis*) in humans [9,10]. Over 1.9 million immunocompromised people in Sub-Saharan Africa frequently develop potentially life-threatening bacteremia from invasive nontyphoidal salmonellosis [11,12].

The burden of *Salmonella* species had different magnitudes in different parts of Ethiopia. For instance, 4% of the 150 stool specimens in Amhara [13]; 2% of among 387 blood and stool specimens in Addis Ababa [14];14% of among 387 blood specimens in Addis Ababa [15]; 3.87% of the 232 stool specimen in Oromia [16]; 7% of among 2000 stool specimen in Jigjiga [17]; and 1.5% of among 381 blood specimens in SNNP [18].

Globally, there has been an increase in multidrug resistance, which is a hazard to public health. The emergence of multidrug-resistant (MDR) bacterial pathogens from diverse sources, such as humans, birds, cattle, and fish, has been the subject of several recent studies. This has increased the need for routine application of antimicrobial susceptibility testing to identify the preferred antibiotic and to screen for the emergence of MDR strains [19–22]. The emergence of MDR and extensively drug-resistant (XDR) *Salmonella* strains has seriously jeopardized the efficacy of several medications, leaving few options for the treatment of both moderate and severe typhoid fever infections. consequently, MDR strains of *S. typhi* are typically thought to be resistant to at least one out of three or more categorically distinct antimicrobials, such as ampicillin, sulfonamides, and chloramphenicol, In contrast, XDR is those that are resistant to all but one or two antimicrobials, showing resistance to many antibiotics including fluoroquinolones, ampicillin, sulfonamides, and third-generation cephalosporin's, and chloramphenicol, leaving out a few effective alternatives for treatment, including piperacillin/tazobactam, azithromycin, and carbapenem [23–25].

*Salmonella* species are on the World Health Organization's list of antibiotic-resistance "priority pathogens" at high threat levels [26]. In Ethiopia, the increasing emergence of antimicrobial-resistant *Salmonella* serovars species to commonly prescribed antimicrobials such as chloramphenicol, ampicillin, tetracycline, and co-trimoxazole has been documented in recent years [27,28]. The burden of multidrug resistance *Salmonella* species had different magnitudes in different parts of Ethiopia. For instance, 66.7% % in Bahir Dar; 25% in Jigjiga [17]; and 83.3% in SNNP [18].

Despite the use of new antibacterial drugs, enteric fevers like typhoid and paratyphoid are caused by multidrug-resistant bacterial strains, which have become one of the main health issues [29]. Knowing the prevalence and multidrug patterns of *Salmonella* species will help to choose the right antimicrobial treatment and reduce the spread of infection. Consequently, the purpose of this study was to evaluate the prevalence, antibiotic resistance, and multidrug resistance patterns of *Salmonella* isolates from various clinical specimens at the University of Gondar Comprehensive Specialized Hospital in Northwestern Ethiopia.

## Methods and materials

### Study design, period, and area

A retrospective hospital-based cross-sectional study was conducted to determine the prevalence, antimicrobial resistance, and multidrug resistance patterns of *Salmonella* isolates among various clinical specimens at the University of Gondar Comprehensive Specialised Hospital (UoGCSH) from June 1st, 2017 to June 3rd, 2022. The study was conducted at the University of Gondar Comprehensive Specialised Hospital in Gondar, Ethiopia. Gondar town is located in the northern part of Ethiopia in Amhara National Regional State, Central Gondar Zone, at a distance of 747km from Addis Ababa and 175 km from Bahir Dar, 12˚ 45˚ north latitude and 370˚ east longitude. Based on the 2016 population estimate, Gondar has a total population of 621,168 with 3200/km2 [30]. Currently, Gondar town has one comprehensive specialized

hospital, which is located in north Ethiopia, and eight government health centers. The hospital has more than 550 beds and provides health services such as surgery, medicine, pathology, TB/HIV, dermatology, antenatal care, laboratory and pharmacy, and maternal and neonatal care for the local community and referred patients from remote areas. The hospital is used as a referral center for more than seven million catchment populations. About more than 12700 patients have visited the outpatient department and more than 3950 visited the inpatient department each month, based on the records of the hospital.

## Study population, sample size, and sampling technique

The source populations were all patients who visited UoG-CSH, and their culture results were registered in the Medical Microbiology Laboratory Logbook. The study population was all patients who had been suspected of having *Salmonella* infection and whose all specimen culture results were registered in the Medical Microbiology Laboratory registration logbook for the last six years. Patients' information, such as socio-demographic characteristics, culture results, and antimicrobial susceptibility test results, which were completely registered in the log book, was included. Patient information with incomplete socio-demographic characteristics, culture results, and antimicrobial susceptibility test results in the log book was excluded.

## Data collection and laboratory methods

### Data collection

Data have been reviewed and then manually collected using a data collection checklist from the registration book at the UoG-CSH in the Medical Microbiology Laboratory section. The patient's socio-demographic characteristics, bacterial isolates, and antimicrobial susceptibility test results were collected using a data collection checklist.

### Laboratory methods

**Isolation and identification of *Salmonella* species.** *Salmonella* species were identified and isolated using the standardized technique from clinical samples [31]. *Salmonella* suspected blood samples were cultured using tryptic soya broth (Oxoid, Ltd. United Kingdom), and other clinical samples were cultured using MacConkey (MAC, Oxoid, Ltd. United Kingdom), and Xylose Lysine Deoxycholate agar (XLD, Oxoid, Ltd. United Kingdom), then incubated at 37˚c for 24 hours. Each of the blood culture vials was monitored for seven consecutive days after its incubation at 37˚C. After seven days, a blood culture vial with no evidence of visible microbial growth was sub-cultured before ruling it to be negative for the patient. For the positive blood culture vials, the samples were cultured onto enriched (blood agar plate and chocolate agar plate, Oxoid, Ltd. United Kingdom), and selective medium (MAC and XLD, Oxoid, Ltd. United Kingdom) for the growth of the suspected pathogen [32]. After overnight incubation at 37˚C, the growth of *Salmonella* species was differentiated and identified by their colony characteristic appearance on XLD agar (red with a black center) and MAC (non-lactose fermenter and colorless colonies) followed by Biochemical tests such as triple sugar iron (Oxoid, Ltd. United Kingdom), Sulfide Indole Motility (Oxoid, Ltd. United Kingdom), Citrate utilization test (Oxoid, Ltd. United Kingdom), lysine decarboxylase (Oxoid, Ltd. United Kingdom) and urease test (Oxoid, Ltd. United Kingdom [33]. Those salmonella that were not identified by conventional biochemicals at the species level were further identified using the API-20E test (BioMerieux SA, France.).The API-20E plastic strip holds twenty mini-test chambers containing dehydrated media with chemically defined compositions for each test. API 20E microtubes were filled up to the edge with the bacterial suspension. A bacterial suspension was used to

rehydrate each of the wells. After incubation, some of the wells developed color changes due to pH differences; others produced end products that had to be identified with reagents. Sterile oil was added into the arabinose, arginine, Adonitola, mannitol, ornithine decarboxylase, Rhamnose, hydrogen sulfide, and urease production test compartments to create anaerobiosis [34]. A profile number was determined from the sequence of positive and negative test results. Then changes were looked up in a codebook to identify the bacterial isolate by using APIWEB [34] (Fig 1).

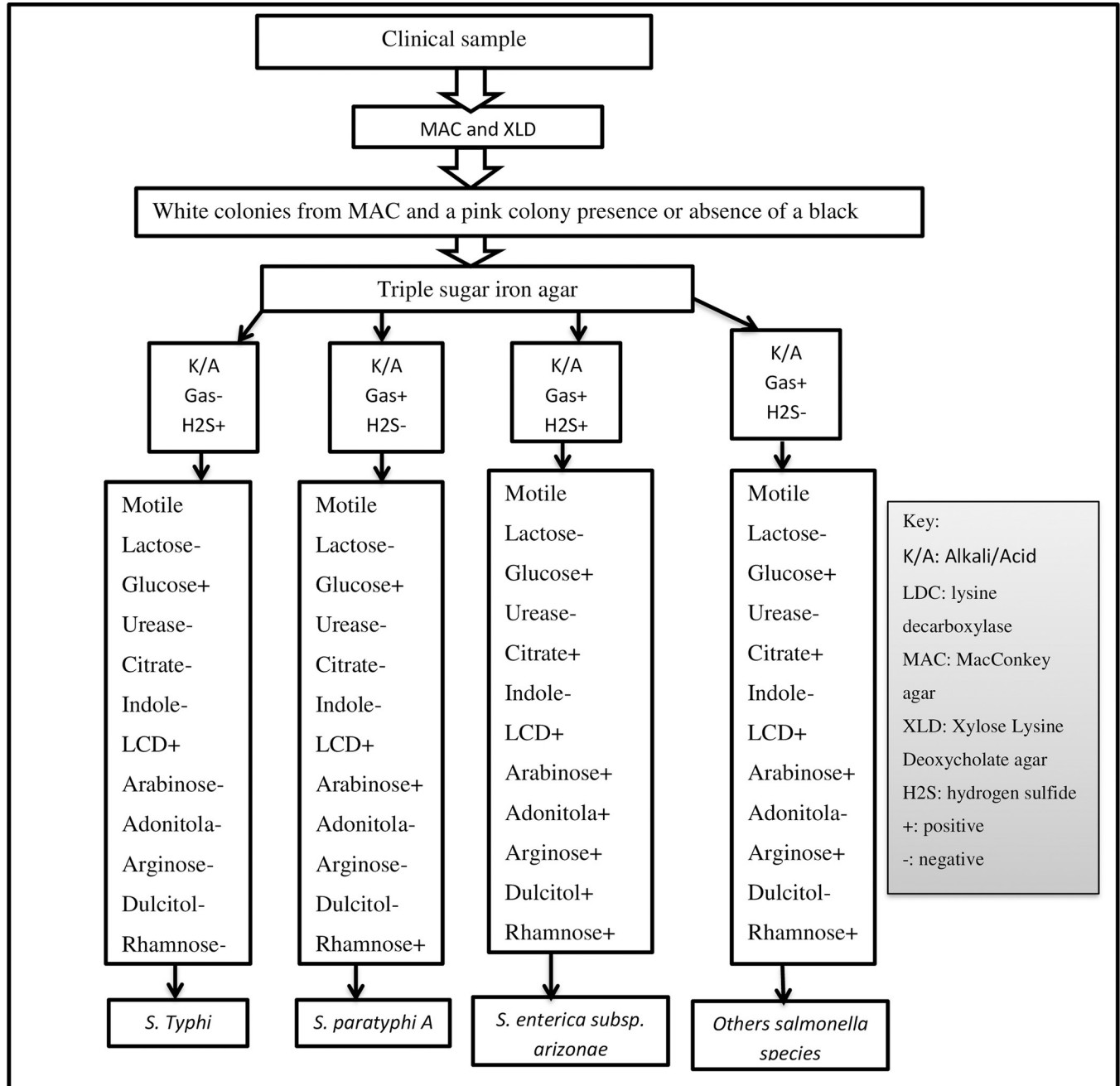

**Fig 1. Flow chart for identification of Salmonella species at UoG-CSH, Gondar, Northwest Ethiopia, from June 2017–June 2022.**

## Antimicrobial susceptibility testing

The Clinical and Laboratory Standard Institute (CLSI) recommendation was followed when doing the antimicrobial susceptibility testing (AST) on the isolates once the bacteria had been identified. This technique was a modified Kirby-Bauer disk diffusion technique. Pure colonies from a young culture were picked and emulsified in 0.85% sterile normal saline to make the bacterial suspension, which was then compared to 0.5 McFarland turbidity standards. Then the bacterial suspension was inoculated onto Muller-Hinton agar (MHA, Oxoid, Ltd. United Kingdom). The following antibiotics disks were tested: ampicillin (10μg), cefotaxime (30μg), ceftazidime (30μg), ceftriaxone (30μg), nalidixic acid (30μg), gentamicin (10μg), meropenem (10μg), tetracycline (30μg), ciprofloxacin (30μg), chloramphenicol (30μg), and trimethoprim/sulphamethoxazole (1.25/23.7μg). These antibiotic discs were from Oxoid, Ltd. United Kingdom. Then the plates were incubated at 37˚C for 24 hours. After overnight incubation, the zone of inhibition was measured and interpreted as susceptible, intermediate, and resistant based on the recommendation of CLSI [35]. Multi-drug resistance patterns of the isolates were identified using the criteria set by Magiorakos et.al. [36].

## Quality control

The quality of data was assured by using a structured data collection format and the format had subcomponents like socio-demographic characteristics, type of specimen collected, ward and type of Salmonella species isolated, and antimicrobial susceptibility pattern results of Salmonella isolates for recommended antibiotics. To avoid the technical errors that would be encountered during data collection, internal quality control has been done by data collector cross-checking. The data collection formats of each data collector were checked daily for completeness of missed or other relevant information to meet supervision during data collection by the principal investigators.

The culture media were checked for sterility by overnight incubation with 5% of the newly prepared media. *E. coli* American type culture collection *(*ATCC) 25922, *P. aeruginosa* ATCC 27853, *S. Typhimurium* ATCC 14028, and *S. flexnerii* ATCC 12022 [37] were reference strains used to check the performance of the culture media, biochemical tests, and antimicrobial disks Moreover, quality control strains were tested for growth, resistance pattern, and biochemical tests in parallel with clinical specimens to assess the validity of the test procedure. The expiration date of all reagents, supplies, and antimicrobial discs was also checked.

## Statistical analysis

Data was entered and analyzed using SPSS version 26. The socio-demographic factors and the prevalence of *Salmonella* infection among the study participants were analyzed using Bivariable logistic regression. To determine factors that are statistically significantly associated with the presence of *Salmonella* infections, variables with p≤ 0.2 in the Bivariable logistic regression analysis have proceeded to the multivariable logistic regression analysis. A p-value of < 0.05 was considered statistically significant at a 95% confidence level. Then the result was presented in the form of a table and figure.

## Ethical approval

We conducted the study following the Declaration of Helsinki. Ethical approval was obtained from the Research and Ethical Review Committee of the School of Biomedical and Laboratory Sciences, College of Medicine and Health Sciences, University of Gondar, with reference number SBLS/354/, date 21/06/2022. All data were fully anonymized before we accessed it, and the

ethics committee waived the requirement for informed consent. We disclosed the purpose of this study to the hospital director and laboratory personnel working in the hospital's bacteriology laboratory, and permission was obtained from the UoG-CSH. In this study, the investigators kept the privacy and confidentiality of the study participants and assured them that it was collected for research purposes only.

## Results

### Socio-demographic features

A total of 26154 recorded patients' data were included in the study. Out of these, 14139 (54%) were males and 12015 (45.9%) were females (**Table 1**). Among the 26,154 different clinical samples sent to the microbiology laboratories during the study period, a total of 41(0.16%) *Salmonella* species were isolated (**Fig 2**).

### Prevalence of *Salmonella* species infection

In this study, most of the identified *Salmonella* species were *S. enterica subsp. arizonae* 21 (51.2%), followed by *S. paratyphi* A 9 (22%), while the remaining 6 (14.6%) were *S. typhi* **(Fig 3)**.

### The distribution of identified *Salmonella* species from socio-demographic features and different clinical samples

Most of the *Salmonella* isolates were found in the age group of less than 19 years 19 (46.3%), followed by the age group 19–44 years 11 (26.8%) (**Table 2**). The majority of *Salmonella* species were isolated from stool 17 (41.5%), followed by blood 9 (22%), urine 8 (19.5%), and body fluid 3 (7.3%). The most prevalent *Salmonella* species isolated in blood culture-positive

**Table 1. Socio-demographic features of study participants at UoG-CSH, Gondar, Northwest Ethiopia, from June 2017–June 2022.**

| Variables | Frequency N (%) | percentage |
|---|---|---|
| **Age group** | | |
| < 6 | 10,704 | 40.9 |
| 6–12 | 1,922 | 7.3 |
| 13–18 | 2,829 | 10.8 |
| 19–44 | 7,528 | 28.8 |
| 45–64 | 3,789 | 14.5 |
| > 64 | 3,474 | 13.3 |
| **Gender** | | |
| Female | 12,015 | 46 |
| Male | 14,139 | 54 |
| **Region** | | |
| Rural | 9,136 | 35 |
| Urban | 17,018 | 65 |
| **Types of wards** | | |
| OPD | 13,232 | 50.6 |
| IPD | 6,142 | 23.5 |
| ICU | 6,771 | 25.9 |
| **Total** | **26,154** | **100** |

Key: OPD: Outpatient department, IPD: Inpatient department, ICU: Intensive care.

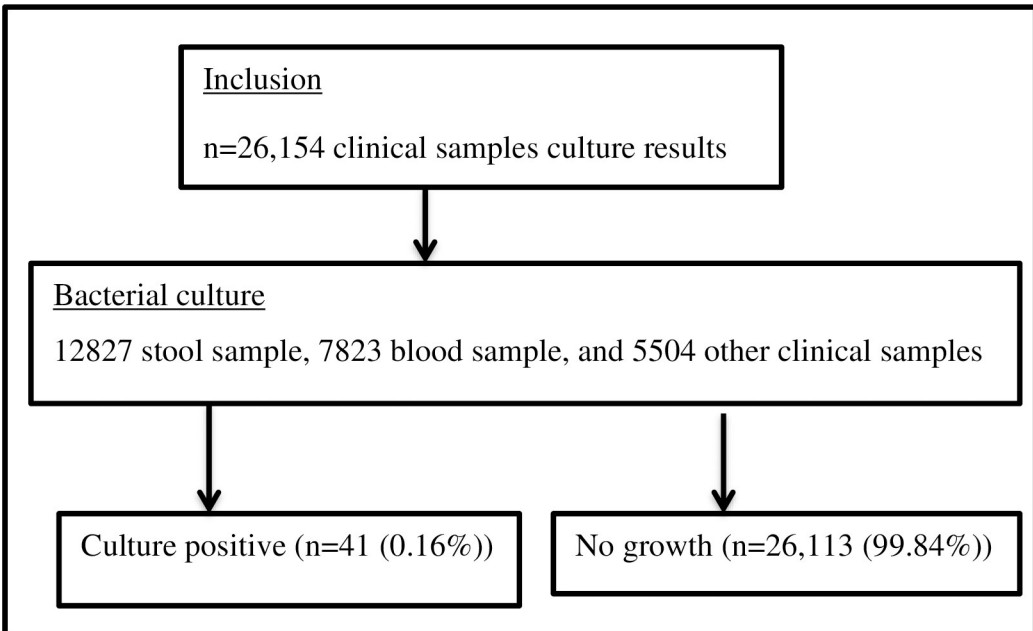

**Fig 2. Flow diagram of participants' inclusion and culture results at UoG-CSH, Gondar, Northwest Ethiopia, from June 2017–June 2022.**

specimens were *S. enterica subsp. arizonae* 9 (22%). The overall distribution of the *Salmonella* isolates from each clinical sample is summarized in **Table 3**.

**The distribution of identified *Salmonella* species by the year.** In this study, the majority (21.9%) of *Salmonella* species were identified by the years 2017 and 2019, while the low (12.2%) numbers of *Salmonella* species were identified by the years 2018 and 2022 (**Table 4**).

## Factors associated with the prevalence of *Salmonella* isolates

Study participants' age, gender, region, and types of wards were selected from bivariate for multivariate analysis (p<0.2). In multivariate analysis, those study participants in the age group of < 6 years (AOR: 7.4, 95% CI: 1.25–43.96) and rural residencies (AOR: 2.07, 95% CI: 1.06–4.03) were found to have high odds of enteric fever (Table 5).

## Antibiotic susceptibility patterns of isolates

The antimicrobial susceptibility patterns of *Salmonella* species were tested against selected antibiotics and the results showed that all isolates were 100% susceptible to cefotaxime, ceftazidime, meropenem, and gentamycin. Of the *Salmonella* isolates, *S. typhi* were 100% resistant to ampicillin, followed by tetracycline (83.3%), trimethoprim/Sulphamethoxazole (83.3%), nalidixic acid (50%), and Ciprofloxacin (16.7%). Furthermore, *S. paratyphi* A was resistant to ampicillin (100%), and 88.9% to tetracycline and chloramphenicol. While *S. enterica subsp. arizonae* were resistant to ampicillin 100% followed by 47.6% to tetracycline and 62% to trimethoprim Sulphamethoxazole (**Table 6**).

## Trends of antibiotic resistance in *Salmonella* isolates

The trend analysis based on the linear regression model showed that the isolates exhibited upward resistance trends to ampicillin, tetracycline, trimethoprim-sulphamethoxazole, and

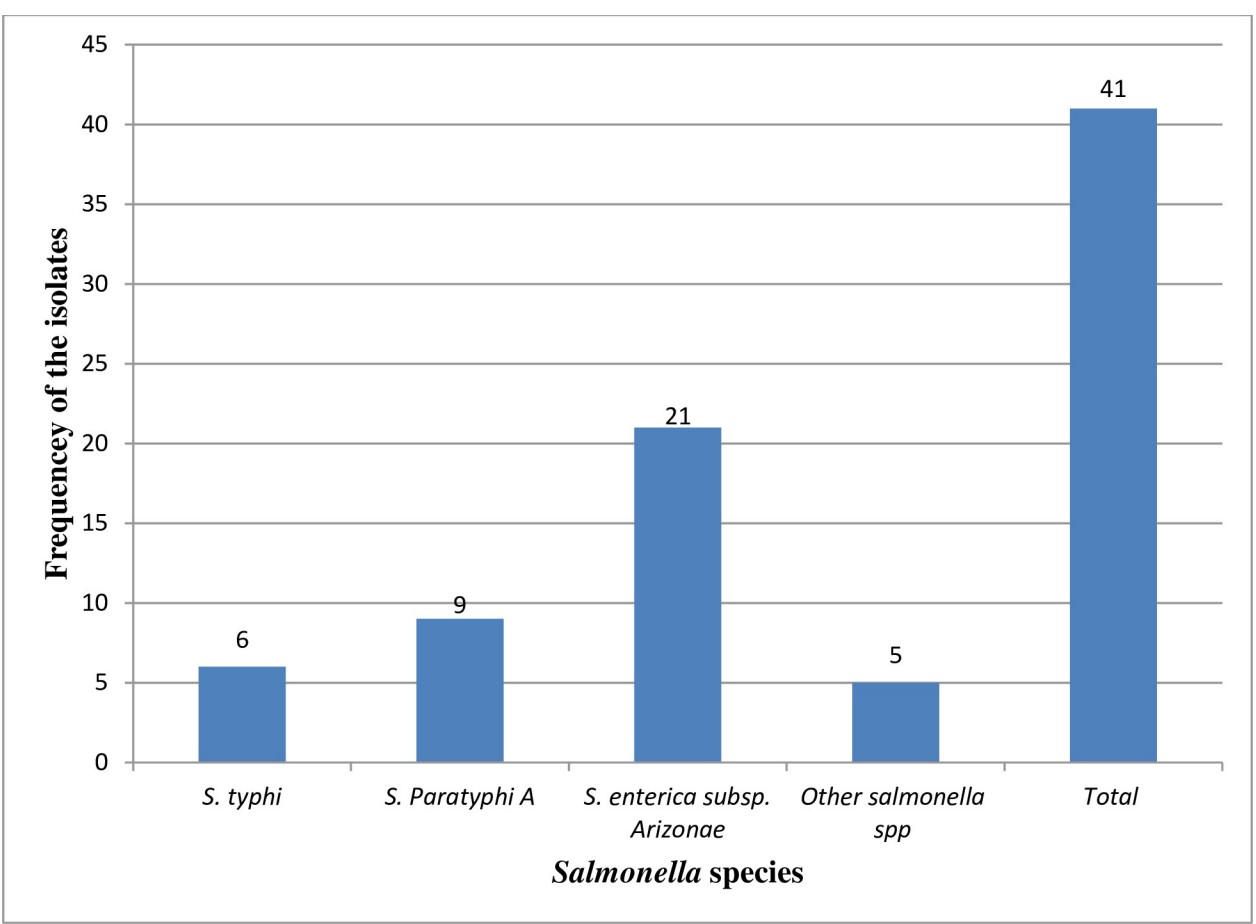

**Fig 3. Frequency of Salmonella species from the study participants at UoG-CSH, Gondar, Northwest Ethiopia, from June 2017-June 2022.**

chloramphenicol with the resistance rates increased by 10.2%, 4.8%, 7%, and 4% for each additional year, respectively. While upward and decreasing resistance rates were observed for nalidixic acid and ceftriaxone for each additional year, respectively (**Fig 4**); however, six-year data could not fit the linear regression model to conduct statistical tests. The trend analysis of MDR *Salmonella* isolates exhibited relatively increased from year to year. The prevalence of MDR was 28.1% in 2017, 44.3% in 2018, 49.8% in 2019, 53.2% in 2020, 54.3% in 2021, and 57.8% in 2022 (**Fig 5**).

## Prevalence of multidrug resistance *Salmonella* species

The overall prevalence of MDR was 53.7% (95% CI: 39.7–61). Accordingly, from the isolates of MDR *Salmonella* species *S. paratyphi* A accounted for the highest 9 (100%), followed by *S. typhi* 66.6% [4], *S. enterica subsp. arizonae* 38.1% [8], and other *Salmonella* species 20% [38] (**Table 7**).

## Discussion

Salmonellosis is one of the main causes of illness and death worldwide, although it is particularly prevalent in developing countries [1,39]. Antimicrobial-resistant *Salmonella* species is a global clinical problem [40]. A pan-resistant *Salmonella* strain was recently reported [40]. The

**Table 2. The distribution of identified *Salmonella* species in socio-demographic features of study participants at UoG-CSH, Gondar, Northwest Ethiopia, from June 2017–June 2022.**

| Variables | Presence of Salmonella isolates | | Identified salmonella isolates (N = 41) | | | |
|---|---|---|---|---|---|---|
| | Yes | No | S. typhi (N = 6) | S. Paratyphi A (N = 9) | S. enterica subsp. arizonae (N = 21) | Other salmonella spp (N = 5) |
| | N (%) | N (%) | N (%) | N (%) | N (%) | N(%) |
| **Age group** | | | | | | |
| < 6 | 19(0.02) | 10,685(99.8) | 5 (12.2) | 5 (12.2) | 8 (19.5) | 1 (2.4) |
| 6–12 | 5 (0.3) | 1,917 (99.7) | 0 | 0 | 5 (12.2) | 0 |
| 13–18 | 3 (0.1) | 2,826 (99.9) | 0 | 1 (2.4) | 1 (2.4) | 1 (2.4) |
| 19–44 | 11 (0.2) | 7,517 (99.8) | 1 (2.4) | 3 (7.3) | 5 (12) | 2 (4.9) |
| 45–64 | 3 (0.1) | 3,786 (99.9) | 0 | 0 | 2 (4.9) | 1 (2.4) |
| > 64 | 0 | 3,474 (100) | 0 | 0 | 0 | 0 |
| **Gender** | | | | | | |
| Female | 17 (0.1) | 11,998 (99.9) | 2 (4.9) | 8 (19.5) | 6 (14.6) | 2 (4.9) |
| Male | 24 (0.2) | 14,115 (99.8) | 4 (9.8) | 1(2.4) | 15 (36.5) | 3 (7.3) |
| **Residence** | | | | | | |
| Rural | 26 (0.3) | 9,110 (99.7) | 4 (9.8) | 6 (14.6) | 12 (29.3) | 4 (9.8) |
| Urban | 15 (0.1) | 17,003 (99.9) | 2 (4.9) | 3 (7.3) | 9 (21.9) | 1(2.4) |
| **Types of wards** | | | | | | |
| OPD | 28 (0.21) | 13,204 (99.79) | 3 (7.3) | 8 (19.5) | 17 (41.5) | 5 (12.2) |
| IPD | 10 (0.16) | 6,132 (99.84) | 1 (2.4) | 1 (2.4) | 3 (7.3) | 0 |
| ICU | 3 (0.1) | 6,768 (99.9) | 1 (2.4) | 0 | 1 (2.4) | 0 |
| **Total** | **41 (0.15)** | **26,113 (99.85)** | **6 (14.6)** | **9 (21.9)** | **21 (51.2)** | **5(12.2)** |

Key: OPD: Outpatient department, IPD: Inpatient department, ICU: Intensive care.

prevalence of multidrug-resistant *Salmonella* has increased, and sub-Saharan Africa has reported outbreaks due to these strains [41]. In developing countries, including Ethiopia, where a high frequency of resistant *Salmonella* species against different antimicrobial agents has been reported and the therapeutic management of the disease is difficult [42,43].

In the present study, out of the total samples positive for *Salmonella* species, 54% were isolated from males and 46% from females which is higher than a study conducted in Nigeria, male 24% and female 17% [44] and in Kathmandu, Nepal, males 5.5% and females 4.2% [45]. This variation may be due to population characteristics, geographic location, participants' hygiene habits, level of education, and the distribution of predisposing factors for bacterial contamination such as consumption of raw meat.

**Table 3. The distribution of identified *Salmonella* species in different clinical samples at UoGCSH, Gondar, Northwest Ethiopia, from June 2017 to June 2022.**

| Clinical isolates | Types of specimens | | | | | | |
|---|---|---|---|---|---|---|---|
| | Stool | Urine | Blood | Discharge | Body fluid | CSF | Total |
| | N (%) | N (%) | N (%) | N (%) | N (%) | N (%) | N (%) |
| *S. typhi* (n = 6) | 2 (4.87) | 1 (2.43) | 2 (4.87) | 0 (0) | 0 (0) | 1 (2.43) | 6 (14.6) |
| *S. paratyphi A* (n = 9) | 4 (9.75) | 3 (7.3) | 2 (4.87) | 0 (0) | 0 (0) | 0 (0) | 9 (22) |
| *S. enterica subsp. arizonae* (n = 21) | 9 (21.95) | 3 (7.3) | 3 (7.3) | 2 (4.87) | 3 (7.3) | 0 (0) | 21(51.2) |
| Other *salmonella* species (n = 5) | 2 (4.87) | 1 (2.43) | 1 (2.43) | 0 (0) | 0 (0) | 1 (2.43) | 5 (12.2) |
| Total (n = 41) | 17 (41.5) | 8 (19.5) | 9 (22) | 2(4.87) | 3 (7.3) | 2 (4.8) | 41 (100) |

Key: CSF: Cerebrospinal fluid.

**Table 4. The distribution of identified *Salmonella* species by the year at UoGCSH, Gondar, Northwest Ethiopia, from June 2017 to June 2022.**

| Year of isolation | Identified salmonella isolates (N = 41) | | | | |
|---|---|---|---|---|---|
| | *S. typhi* (N = 6) | *S. Paratyphi* A (N = 9) | *S. enterica subsp. Arizonae* (N = 21) | Other *Salmonella* species (N = 5) | Total (N = 41) |
| | N (%) | N (%) | N (%) | N (%) | N (%) |
| 2017 | 2 (4.9) | 1(2.4) | 4 (9.8) | 2 (4.9) | 9 (21.9) |
| 2018 | 1 (2.4) | 1 (2.4) | 2 (4.9) | 1(2.4) | 5(12.2) |
| 2019 | 0 | 3 (7.3) | 6 (14.6) | 0 | 9 (21.9) |
| 2020 | 1(2.4) | 1(2.4) | 3 (7.3) | 2 (4.9) | 7 (17) |
| 2021 | 0 | 2 (4.9) | 4 (9.8) | 1(2.4) | 7 (17) |
| 2022 | 2 (4.9) | 1 (2.4) | 2 (4.9) | 0 | 5(12.2) |

In the current study, the overall prevalence of *Salmonella* species was 0.15%. This was comparable with previous reports in India 0.53% [46] and in Fijji 0.7% [47]. Besides, this study is lower than a study conducted in Ethiopia: Addis Ababa (4.1%) [48], in Ethiopia: Adigrat (7.3%) [49], Yemen 20% [50], in Nepal (3.1%) [51], and in Bangladesh (5%) [52]. The discrepancies in all of the above comparisons could be attributed to the fact that different, countries and within countries may be due to applying different test methods for the isolation of *Salmonel*la species, economic status, characteristics of study participants, and epidemiological distributions of the pathogenic bacteria.

In the present study the identification of S*almonella* species from stool samples and blood samples was comparable to reports was Ethiopia [14]. Besides, this report is lower than that of a study conducted in Ethiopia [3,53] and in Yemen [50]. The discrepancy may be partly

**Table 5. Factors associated with the prevalence of *Salmonella* isolates among study participants at UoGCSH, Gondar, Northwest Ethiopia, from June 2017 to June 2022.**

| Variables | Presence of *Salmonella* isolates | | COR (95% CI) | p-value | AOR (95% CI) | P-value |
|---|---|---|---|---|---|---|
| | Yes N (%) | No N (%) | | | | |
| **Age group** | | | | | | |
| < 6 | 19 (0.02) | 10,685 (99.8) | 3.96(2.45–14.02) | 0.001 | 7.42(1.25–43.96) | 0.027* |
| 6–12 | 5 (0.3) | 1,917 (99.7) | 1.29(0.24–3.26) | 0.752 | NA | |
| 13–18 | 3 (0.1) | 2,826 (99.9) | 1.46(0.41–6.24) | 0.505 | NA | |
| 19–44 | 11 (0.2) | 7,517 (99.8) | 1.42 (0.67–3.00) | 0.359 | NA | |
| 45–64 | 3 (0.1) | 3,786 (99.9) | 1.07 (.633–1.79) | 0.811 | NA | |
| > 64 | 0 | 3,474 (100) | 1 | | | |
| **Gender** | | | | | | |
| Female | 17 (0.1) | 11,998 (99.9) | 1 | | 1 | |
| Male | 24 (0.2) | 14,115 (99.8) | 1.73 (0.98–3.04) | 0.002 | 1.08 (0.53–2.19) | 0.822 |
| **Residence** | | | | | | |
| Rural | 26 (0.3) | 9,110 (99.7) | 1.40 (0.79–2.47) | 0.001 | 2.07 (1.06–4.03) | 0.032* |
| Urban | 15 (0.1) | 17,003 (99.9) | 1 | | 1 | |
| **Types of wards** | | | | | | |
| OPD | 28 (0.2) | 13,204 (99.8) | 2.50 (1.42–4.40) | 0.001 | 1.58 (0.76–3.29) | 0.219 |
| IPD | 10 (0.2) | 6,132 (99.8) | 1.89 (0.98–3.63) | 0.022 | 1.77(0.86–3.63) | 0.119 |
| ICU | 3(0.1) | 6,768 (99.9) | 1 | | 1 | |

Key: OPD: Outpatient department, IPD: Inpatient department, ICU: Intensive care.

**Table 6. Antimicrobial resistance patterns of *Salmonella* species at UoGCSH, Gondar, Northwest Ethiopia, from June 2017 to June 2022.**

| Antibiotics | Salmonella isolates (N = 41) | | | | | | | |
|---|---|---|---|---|---|---|---|---|
| | *S. typhi* (n = 6) | | *S. para typhi A* (n = 9) | | *S. enterica subsp. arizonae* (n = 21) | | Other *salmonella* species (n = 5) | |
| | S | R | S | R | S | R | S | R |
| | N (%) | N (%) | N (%) | N (%) | N (%) | N (%) | N (%) | N (%) |
| CIP | 5 (83.3) | 1 (16.7) | 3 (50) | 3 (50) | 12 (57.1) | 9 (42.9) | 3 (60) | 2 (40) |
| CRO | 6 (100) | 0 | 4 (44.4) | 5 (55.6) | 15 (71.4) | 6 (.28.6) | 5 (100) | 0 |
| CAZ | 6 (100) | 0 | 6 (100) | 0 | 21 (100) | 0 (0) | 65 (100) | 0 |
| CTX | 6 (100) | 0 | 06 (100) | 0 | 21 (100) | 0 | 5 (100) | 0 |
| NAL | 3 (50) | 3 (50) | 4 (44.4) | 5 (55.6) | 16 (76.2) | 5 (23.8) | 4 (80) | 1 (20) |
| MER | 6 (100) | 0 | 8 (100) | 0 | 8 (100) | 0 | 3 (100) | 0 |
| SXT | 11 (6.7) | 5 (83.3) | 1 (33.3) | 6 (66.7) | 8 (38) | 13 (62) | 1 (20) | 4 (80) |
| TET | 1 (16.7) | 5 (83.3) | 1 (11.1) | 7 (88.9) | 11 (52.3) | 10 (47.6) | 2 (40) | 3 (60) |
| GEN | 5 (100) | 0 | 4 (100) | 0 | 18 (100) | 0 | 4 (100) | 0 |
| CHL | 1 (16.7) | 5 (83.3) | 1 (11.1%) | 8 (88.9) | 15 (71.4) | 6 (28.6) | 3 (60) | 2 (40) |
| AMP | 0 | 6 (100) | 0 | 9 (100) | 0 | 19 (100) | 1 (20) | 4 (80) |

**Key**; S = susceptible, R = resistance, CIP = Ciprofloxaciline, CRO = Ceftriaxone, CAZ = Ceftazidime, CTX = Cefotaxime, MER = Meropenem,

SXT = Sulphamethoxazole-trimethoprim, TET = Tetracycline, GEN = Gentamycin, CHL = Chloramphenicol, NAL = Nalidixic acid, AMP = Ampicillin.

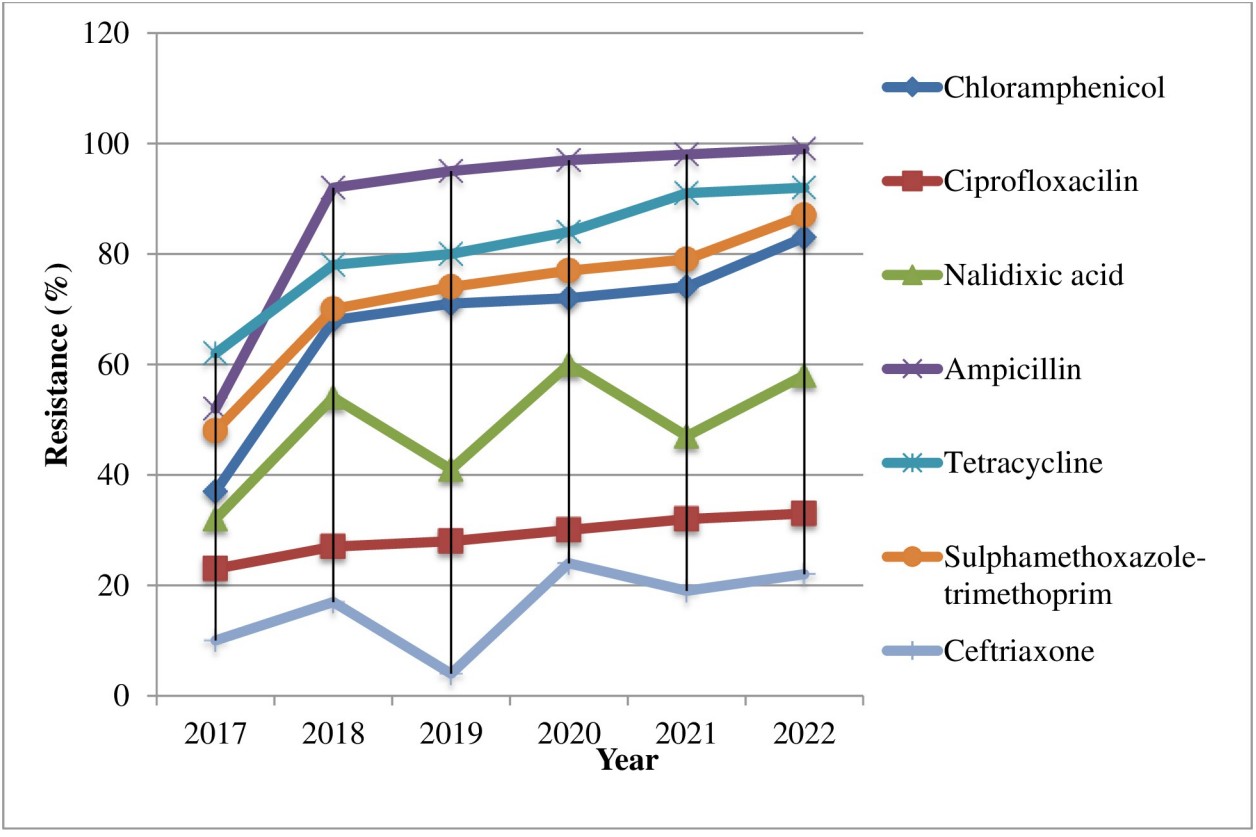

**Fig 4. Trends of Antibiotic Resistance in *Salmonella* isolates at UoG-CSH, Gondar, Northwest Ethiopia, from June 2017–June 2022.**

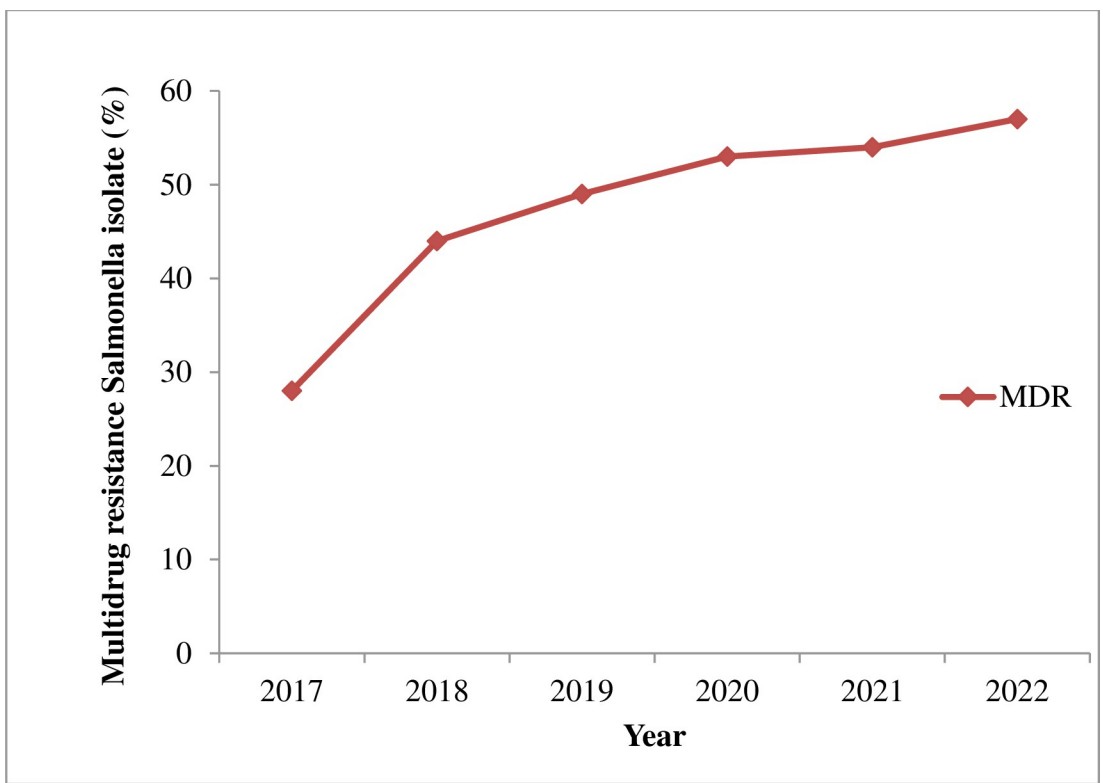

**Fig 5. Trends Multidrug resistance *Salmonella* isolate at UoG-CSH, Gondar, Northwest Ethiopia, from June 2017–June 2022.**

explained by variations in study sites, isolation methods, the studies conducted at different places with different populations, and sample sizes across studies.

Nowadays, *S. typhi* and *S. paratyphi* A antimicrobial resistance becomes worse in low and medium-income countries [54]. Even if AMR surveillance has been established in Ethiopia [55], It is becoming more difficult in areas where self-medication with incorrect antibiotics is widespread and MDR is extremely prevalent [38]. In the current study, *S. typhi* and *S. paratyphi* A displayed varying levels of resistance to the several tested antibiotic classes. The resistance to tetracycline and chloramphenicol was highest in *S. typhi* and *S. paratyphi* A, with

**Table 7. Multidrug resistance profiles of *Salmonella* species in different clinical specimens at UoGCSH, Gondar, Northwest Ethiopia, from June 2017-June 2022.**

| Isolates N = (41) | Level of resistance n(%) | | | | | | | |
|---|---|---|---|---|---|---|---|---|
| | R0 | R1 | R2 | R3 | R4 | R5 | R6 | Total MDR R ≥ R3 |
| | N (%) | N (%) | N (%) | N (%) | N (%) | N (%) | N (%) | N (%) |
| *S. typhi* (n = 6) | 0 | 1 (16.7) | 1(16.7) | 1(16.7) | 2(33.3) | 1 (16.7) | 0 | 4 (66.6) |
| *S. paratyphi* A (n = 9) | 0 | 0 | 0 | 6(66.7) | 1(11.1) | 2(22.2) | 0 | 9 (100) |
| *S. enterica subsp. arizonae* (n = 21) | 3 (14.3) | 5 (23.8) | 5 (23.8) | 4 (19) | 2(9.5) | 1(4.7) | 1(4.7) | 8 (38.1) |
| Other *Salmonella* species (n = 5) | 1 (20) | 2 (40) | 1(20) | 0 | 1(20) | 0 | 0 | 1 (20) |
| Total (N = 41) | 4 (9.8) | 10 (24.4) | 9(22) | 11(26.8) | 6(14.6) | 4 (9.8) | 1(2.4) | 22 (53.7) |

Abbreviations: MDR: Multi-drug resistant, R0: No antibiotic resistance, R1: Resistance to one class, R2: Resistance to two classes, R3: Resistance to three classes, R4: Resistance to four classes, R5: Resistance to five classes, and R6: Resistance to six antimicrobials in different classes.

more than 83% for each. In the current study, the level of chloramphenicol resistance in both of the isolated bacteria was comparable to studies conducted in Bahir Dar–Ethiopia [13], Jigjiga-Ethiopia [17], and Nepal [56]. However, higher than previous studies in Nigeria [57] and India [58]. This might be due to the study period and the geographic location being different. In addition, there has been an increase in antibiotic resistance over time, particularly for tetracycline and chloramphenicol [58].

In the current study, nalidixic acid demonstrated potent against half of *S. typhi* and *S. Paratyphi* A isolates. The resistance level of *S. typhi* against Nalidixic acid in the present study was comparable with a study in India [58]. However, there was a study that reported a higher resistance level [56], and many previous studies documented a lower resistance level of *S. typhi* against Nalidixic acid than the present study [17,52]. Additionally, 83.3% of *S. typhi* and 50% of *S. paratyphi* A isolates were susceptible to ciprofloxacin. In the present study, the resistance level of *S. typhi* against ciprofloxacin was comparable with previous studies [17,52,59]. Besides, a higher resistance level of *S. paratyphi* A was documented than in previous studies [17,52]. The resistance level of ciprofloxacin for both bacteria in the present study was higher than in studies in Nepal [56], Nigeria [57], and India [58]. This variation may be due to geographic location, miss use of antibiotics, and the high resistance level might be because these antimicrobials are most commonly consumed by the community without a prescription as they are easily accessible and relatively inexpensive, and in most public and private healthcare settings, empirical treatment is widely practiced in Ethiopia, resulting in high resistance [60].

On the other hand, cephalosporins such as cefotaxime, cefoxitin, and ceftazidime were effective against all isolates of *S. typhi* and *S. paratyphi* A. Half of *S. paratyphi* A had ceftriaxone resistance, in contrast to S. typhi, which revealed 100% susceptibility. This study is comparable to S. typhi susceptibility results reported for ceftriaxone and ceftazidime in previous studies [13,52,57,58].

The resistance level in the present study was higher than previous study reports for *S. paratyphi* A against ceftriaxone [17,38,52,57,58]. However, contrary to previous findings, *S. paratyphi* A did not show any ceftazidime resistance [57]. The increased resistance of *S. typhi* and *S. paratyphi* A in the current study to various antibiotic classes could be attributed to improper prescription of antibiotics by medical professionals, misuse of antibiotics because antimicrobial resistance varies greatly over time, and gene transfer among different *Salmonella* species [13].

In the current study, all isolates of *S. paratyphi* A and 66.6% of *S. typhi* were found to be MDR. Even though the MDR level of *S. typhi* in the current investigation was comparable to reports in Bangladesh (64.28%) [59], it was higher than studies in Ethiopia [17], Pakistan, and in Nigeria [25,57,58], which reported 0 to 29%. Similarly, the MDR level of *S. paratyphi* A in the present study was higher than studies in Ethiopia [17] and abroad [54,57,58] which documented 0 to 25%. The high MDR level was supported by a meta-analysis study that showed an escalating AMR trend among *S. typhi* and *S. paratyphi* A [54].

## Limitations of the study

Since the current study is retrospective it is difficult to know the clinical risk factors associated with multidrug resistance strains of *Salmonella* isolates.

## Conclusions and recommendations

A low prevalence of *Salmonella* species was observed in six years. However, *S. typhi* and *S. paratyphi* A showed higher resistance rates than previously reported to routinely administered antibiotics in the research area, such as ciprofloxacin and ceftriaxone. In addition, all isolates

of *S. paratyphi* A and the majority of *S. typhi* were MDR. On the other hand, cefotaxime and ceftazidime were completely effective against all isolates of *S. typhi* and *S. paratyphi* A. It is strongly recommended to do a thorough nationwide investigation on the disease's prevalence, circulating serotypes' antibiotic susceptibility patterns, as well as the most effective diagnostic techniques. Further large-scale and molecular researches are advised to determine the incidence of *S. typhi* and *S. paratyphi* A and their AMR profiles to identify AMR genes.

## Acknowledgments

The authors would like to thank the Department of Medical Microbiology, School of Biomedical and Laboratory Sciences, College of Medicine and Health Sciences, and the University of Gondar and we also acknowledge study participants.

## Author Contributions

**Conceptualization:** Azanaw Amare, Muluneh Assefa, Alem Getaneh.

**Data curation:** Azanaw Amare, Fekadu Asnakew, Yonas Asressie, Eshetie Guadie, Addisu Tirusew, Silenat Muluneh, Abebew Awoke, Muluneh Assefa, Worku Ferede.

**Formal analysis:** Azanaw Amare, Fekadu Asnakew, Silenat Muluneh, Muluneh Assefa, Alem Getaneh, Mulualem Lemma.

**Investigation:** Azanaw Amare, Fekadu Asnakew, Silenat Muluneh.

**Methodology:** Azanaw Amare, Yonas Asressie, Eshetie Guadie, Addisu Tirusew, Muluneh Assefa.

**Software:** Fekadu Asnakew, Yonas Asressie, Eshetie Guadie.

**Supervision:** Azanaw Amare, Worku Ferede, Alem Getaneh, Mulualem Lemma.

**Validation:** Azanaw Amare, Alem Getaneh.

**Visualization:** Azanaw Amare.

**Writing – original draft:** Azanaw Amare, Fekadu Asnakew, Yonas Asressie, Eshetie Guadie, Addisu Tirusew, Silenat Muluneh, Abebew Awoke, Muluneh Assefa, Worku Ferede, Alem Getaneh, Mulualem Lemma.

**Writing – review & editing:** Azanaw Amare, Abebew Awoke, Muluneh Assefa, Alem Getaneh, Mulualem Lemma.

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
