## [Decision Letter · Decision Letter 0]

9 Oct 2023

PONE-D-23-08664Prevalence of Multidrug Resistance Salmonella species isolated from clinical specimens at University of Gondar Comprehensive Specialized Hospital Northwest Ethiopia: A Retrospective studyPLOS ONE

Dear Amare,

Thank you for submitting your manuscript to PLOS ONE. After careful consideration, we feel that it has merit but does not fully meet PLOS ONE’s publication criteria as it currently stands. Therefore, we invite you to submit a revised version of the manuscript that addresses the points raised during the review process.

**Abstract:**

Make sure that the timeline for the research is reported consistently. It reads June 2017 to June 2017 in some lines. In addition, the abstract must show the methods used to identify Salmonella to the species level and the methods applied for the antimicrobial sensitivity measurements and interpretations. Further, you must clarify how the MDR was determined (one line in the abstract).

**Introduction:**

The introduction is well documented, but it lacks a focus on Ethiopia. Other studies have explored the prevalence of Salmonella spp. in clinical settings in Ethiopia and AMR. The authors should briefly mention what is already known and how the current research enhances the topic's understanding (contribute insights).

**Methods:**

The main concern is identifying Salmonella at the species level, relying solely on a few biochemical tests. While biochemical systems such as API 20E could be used to identify Salmonella at the species level, the methods of choice are molecular. The authors should explain how they determined the species using only urease, TSI, indole motility, and lysine decarboxylase. Could the authors provide the final biochemical profiles for each identified species?

The Salmonella were retrieved from fecal, urine, or CSF samples. It’s unclear if the samples were from the same or different patients. The authors must clarify this.

For antimicrobial susceptibility testing, what were the quality controls in place? How were the intermediates isolates interpreted? Or were there no intermediates isolates?

**Ethical consideration:**

Could the authors provide the ethics approval letter or unique identifier for the IRB approval?

**Results:**

The results make it unclear how many samples were sent to the microbiological laboratory and how many were Salmonella-suspected. Moreover, it’s unclear how the identified isolates were distributed in 2017-2022. Could the authors make a table showing the demographics of the patients? The table should show the age, gender, region, and clinical symptoms of the patients (as this could support an understanding of the low detection level of Salmonella).

Combining the demographics with the isolation/identification results makes reading difficult.

Authors should revise the table titles by making them brief and descriptive.

**Discussion:**

The discussion could be improved. The authors attempted to compare their findings to similar studies, but the interpretation of discrepancies is lacking. The authors often justify the discrepancies in methods used, region, etc. The authors should, for example, go further and explain why the methods used in this study (or the other compared studies) are not optimal for identifying Salmonella.

**Study Limitations:**

There are many limitations to the current study. The significant limitations which reduce the reliability of the findings are:

Methods used for speciation of Salmonella.Antibiotic Sensitivity testing that lacked standardization and quality control

**Data availability:**

Datasets must be available as per PLOS ONE policy.

**General:**

There are many typos which reduce the quality of the manuscript. It’s recommended to proofread the manuscript before resubmission.

We look forward to receiving your revised manuscript.

Kind regards,

Anselme Shyaka, Ph.D

Academic Editor

PLOS ONE

Journal Requirements:

4. Please include a caption for figure 1.

Additional Editor Comments (if provided):

Dear corresponding author,

I am pleased to invite you to review the issues identified in your manuscript and submit a revision, which reviewers will again evaluate.

Reviewers' comments:

Reviewer's Responses to Questions

**Comments to the Author**

1. Is the manuscript technically sound, and do the data support the conclusions?

Reviewer #1: Partly

2. Has the statistical analysis been performed appropriately and rigorously? 

Reviewer #1: N/A

3. Have the authors made all data underlying the findings in their manuscript fully available?

Reviewer #1: No

4. Is the manuscript presented in an intelligible fashion and written in standard English?

Reviewer #1: Yes

5. Review Comments to the Author

Reviewer #1: Review comments

The authors have addressed a very important microbiological topic on antimicrobial resistance and pathogen profiles. Although it is retrospective, it offers enrichment to our current understanding of salmonellosis in the African context.

Abstract:

This is well written but requires editing to improve on the gramma.

Introduction

This chapter should be re-written.

Firstly, the statement Genus Salmonella is the family Enterobacteriaceae which is a Gram-negative, non-lactose ferment is not clear what exactly the author is trying to say.

Readers would be more interested in seeing the burden of salmonellosis across the globe, what is being done and the gaps. The authors should limit the characterization of the organisms at this stage as much of this is already documented elsewhere. Some grammatic errors also need to be corrected including the naming of drugs like naldixic acid, instead of nalidixic acid. This should be corrected throughout the document.

Methods and materials

The author should give us a feel of the hospital capacity eg: bedding, how many patients are seen in OPD and IPD in a month. How many of these have symptoms related to salmonellosis. This will guide us on the generalizability of the findings in this context.

Study population, sample size, and sampling technique

The authors repeatedly describe 2017-2022 as 6 years. This should be revised.

Quality control

The authors should include the QC and QA aspects that were done during culture and AST in the samples that were previously analyzed. Also, give a clear description of the exact QC that was done within your data collection. It is not clear what was checked by data collectors and how.

Results

The authors could include a figure of a study profile that would be helpful here.

Discussion

In the first paragraph, the authors describe salmonella prevalence figures in males and females. Ideally, these should add up to 100% but this is not the case. This should be explained.

In the fourth paragraph, the author should mention the countries in the narrative where the rates are lower or higher than what was observed in this study.

6. PLOS authors have the option to publish the peer review history of their article (what does this mean?). If published, this will include your full peer review and any attached files.

Reviewer #1: **Yes: **Patrick Orikiriza

---

## [Author Response · Author response to Decision Letter 0]

30 Nov 2023

Review PLOS One

Response letter for manuscript (PONE-D-23-08664) 

To Academic Editors and reviewers:

Dear Academic Editor and reviewers,

Authors: We the authors would like to thank the editorial teams of PLOS ONE for quick review process of our manuscript titled “Prevalence of Multidrug Resistance Salmonella species isolated from clinical specimens at University of Gondar Comprehensive Specialized Hospital Northwest Ethiopia”. We appreciate for spending your precious time and forwarding your valuable comments, which have significantly improved our application. We are also grateful for this positive feedback. Please see below, bold, for a point-by-point response to the reviewers. All page numbers refer to the revised manuscript file with tracked changes. All modifications in the manuscript have been highlighted in red.

We are looking forward to hearing from you in due course,

Sincerely,

Azanaw Amare (Corresponding author)

Responses to Editor’s comments

Abstract:

Editor: Make sure that the timeline for the research is reported consistently. It reads June 2017 to June 2017 in some lines. In addition, the abstract must show the methods used to identify Salmonella to the species level and the methods applied for the antimicrobial sensitivity measurements and interpretations. Further, you must clarify how the MDR was determined (one line in the abstract).

Response: Thank you for this suggestion. It would have been interesting to explore this aspect. We tried to add the following suggestion on (page 2, Line: 38-43).

Introduction:

Editor: The introduction is well documented, but it lacks a focus on Ethiopia. Other studies have explored the prevalence of Salmonella spp. in clinical settings in Ethiopia and AMR. The authors should briefly mention what is already known and how the current research enhances the topic's understanding (contribute insights).

Response: We agree with this and have incorporated your suggestions and comments on (page 5, line: 97-104 and page 6, line: 120-125).

Methods:

Editor: The main concern is identifying Salmonella at the species level, relying solely on a few biochemical tests. While biochemical systems such as API 20E could be used to identify Salmonella at the species level, the methods of choice are molecular. The authors should explain how they determined the species using only urease, TSI, indole motility, and lysine decarboxylase. Could the authors provide the final biochemical profiles for each identified species?

Response: We agree with this and have incorporated your comments on (page 8, line: 176 -183). However molecular methods were taken as limitations (page 19, Line: 370–372).

Editor: The Salmonella were retrieved from fecal, urine, or CSF samples. It’s unclear if the samples were from the same or different patients. The authors must clarify this.

Response: We are also grateful for the suggestion and thank you. The Salmonella were retrieved either from fecal, urine, or CSF samples from different patients. 

Editor: For antimicrobial susceptibility testing, what were the quality controls in place? How were the intermediates isolates interpreted? Or were there no intermediates isolates?

Response: Thank you for bringing this issue to our intention. We incorporate quality controls for antimicrobial susceptibility testing on (page 10, line: 208-214). However we did not found intermediate results during data collection. 

Editor Ethical consideration: Could the authors provide the ethics approval letter or unique identifier for the IRB approval?

Response: Thank you for your comment. We have incorporated unique identifier for the ethical approval on (page 10, line: 223).

Editor: The results make it unclear how many samples were sent to the microbiological laboratory and how many were Salmonella-suspected. Moreover, it’s unclear how the identified isolates were distributed in 2017-2022.

Response: thank you for your direction. We tried to clarify using text and tables on (page 14)

 Editor: Could the authors make a table showing the demographics of the patients? The table should show the age, gender, region, and clinical symptoms of the patients (as this could support an understanding of the low detection level of Salmonella). Combining the demographics with the isolation/identification results makes reading difficult.

Response: thank you for your direction. We tried to clarify the demographics of the patients (age, gender, and region) using tables on (page 12). 

Editor: Authors should revise the table titles by making them brief and descriptive.

Response: Thank you for reflecting on your concern. We amended and made the table titles brief and descriptive through the manuscript.

Editor: The discussion could be improved. The authors attempted to compare their findings to similar studies, but the interpretation of discrepancies is lacking. The authors often justify the discrepancies in methods used, region, etc. The authors should, for example, go further and explain why the methods used in this study (or the other compared studies) are not optimal for identifying Salmonella.

Response: Thank you for your input. We tried to address all your concerns in the revised manuscript on (page 17: line 296-298, line 303-305, and line 309-311, page 18:line 314-316, line 325-327, and line 332-334 and page 19:line 344-348 and line 355-39) .

Editor: There are many limitations to the current study. The significant limitations which reduce the reliability of the findings are:

1. Methods used for speciation of Salmonella.

2. Antibiotic Sensitivity testing that lacked standardization and quality control

Response: Thank you for pointing out these potential limitations. We included this information in the limitation part on (page 19, Line: 370–372).

Editor: Datasets must be available as per PLOS ONE policy.

Response: authors to make all data necessary publicly available without restriction at the time of publication.

Response: We believed that we are following PLOS ONE’s style requirements

Editor: There are many typos which reduce the quality of the manuscript. It’s recommended to proofread the manuscript before resubmission.

Response: Thank you for your input. We attempted to proofread review the entire manuscript

Response: Thank you. We prepare and upload a response letter for each editor and reviewer(s) comments

Response: We prepared a track change file that highlights changes that we made to the original version during revision.

Response: We prepared a revised manuscript without truck change and uploaded as 'Manuscript' in the submission system

Response: Thank you for your input. We updated the financial disclosure in the cover letter. 

If applicable, we recommend that you deposit your laboratory protocols in protocols.io to enhance the reproducibility of your results. Protocols.io assigns your protocol its own identifier (DOI) so that it can be cited independently in the future. For instructions see: https://journals.plos.org/plosone/s/submission-guidelines#loc-laboratory-protocols. Additionally, PLOS ONE offers an option for publishing peer-reviewed Lab Protocol articles, which describe protocols hosted on protocols.io. Read more information on sharing protocols at https://plos.org/protocols?utm_medium=editorial-email&utm_source=authorletters&utm_campaign=protocols

Response: NA

Journal Requirements:

Comments: 1. please ensure that your manuscript meets PLOS ONE's style requirements, including those for file naming. The PLOS ONE style templates can be found at

Response: During revision of our manuscript, we tried to follow the requirements of the journal PLOS ONE's style

Comments: 2. please provide additional details regarding participant consent. In the ethics statement in the Methods and online submission information, please ensure that you have specified what type you obtained (for instance, written or verbal, and if verbal, how it was documented and witnessed). If your study included minors, state whether you obtained consent from parents or guardians. If the need for consent was waived by the ethics committee, please include this information.

Response: Thank you for your comment. The participant consent was waived (obtained, documented and witnessed) by the ethics committee before we accessed the data. Please see it on page 10: line 223-227. 

Comments: 3. we note that you have stated that you will provide repository information for your data at acceptance. Should your manuscript be accepted for publication, we will hold it until you provide the relevant accession numbers or DOIs necessary to access your data. If you wish to make changes to your Data Availability statement, please describe these changes in your cover letter and we will update your Data Availability statement to reflect the information you provide.

Response: Authors: Thank you. Please update it

Comments: 4. please include a caption for figure 1.

Response: Thank you for your input. We include the figure in caption.

Responses to reviewers' comments

Reviewer #1: Review comments

 Comments: the authors have addressed a very important microbiological topic on antimicrobial resistance and pathogen profiles. Although it is retrospective, it offers enrichment to our current understanding of salmonellosis in the African context.

Response: We appreciate you taking the time to thoroughly review this manuscript and provide us with constructive feedback, which we believe improved the overall quality of our paper. We made every effort to address all your concerns. We've copied your comments and responses below to make things easier for you.

Abstract:

Comments: This is well written but requires editing to improve on the gramma.

Response: We are grateful for the suggestion and thank you. We went throughout the entire manuscript to eliminate grammatical and editing mistakes.

Introduction

Comments: This chapter should be re-written.

Firstly, the statement Genus Salmonella is the family Enterobacteriaceae which is a Gram-negative, non-lactose ferment is not clear what exactly the author is trying to say.

Readers would be more interested in seeing the burden of salmonellosis across the globe, what is being done and the gaps. The authors should limit the characterization of the organisms at this stage as much of this is already documented elsewhere.

Response: thank you very much for your direction, suggestion, and comments. We agree with this and have incorporated your suggestion in the revised manuscript on (page 3:line 64-65 and page 4: line 69-77).

 Comments: Some grammatic errors also need to be corrected including the naming of drugs like naldixic acid, instead of nalidixic acid. This should be corrected throughout the document.

Response: Response: thank you very much for your direction, suggestion, and comments. We went throughout the entire manuscript to eliminate grammatical and editing mistakes.

Methods and materials

Comments: The author should give us a feel of the hospital capacity eg: bedding, how many patients are seen in OPD and IPD in a month. How many of these have symptoms related to salmonellosis. This will guide us on the generalizability of the findings in this context.

Response: Thank you for this suggestion. It would have been interesting to explore this aspect. We tried to add the following suggestion on (page 7: Line 140-145).

Comments: Study population, sample size, and sampling technique

the authors repeatedly describe 2017-2022 as 6 years. This should be revised.

Response: Thanks for your kind reminders. We revised it accordingly. 

Quality control

Comments: The authors should include the QC and QA aspects that were done during culture and AST in the samples that were previously analyzed. Also, give a clear description of the exact QC that was done within your data collection. It is not clear what was checked by data collectors and how.

Response: thank you very much for your suggestions; the data collector were checked QC and QA in in records of QC and QA that was done by Laboratory technologist and we incorporate on (page 10 :line 208-214)

Results

Comments: The authors could include a figure of a study profile that would be helpful here.

Response: Thank you for your valuable suggestion and we incorporate in figure 1 Discussion

Comments: In the first paragraph, the authors describe salmonella prevalence figures in males and females. Ideally, these should add up to 100% but this is not the case. This should be explained.

Response: Thank you for this suggestion. We went throughout the entire manuscript to eliminate editing mistakes.

Comments: In the fourth paragraph, the author should mention the countries in the narrative where the rates are lower or higher than what was observed in this study.

Response: Thank you for your input. We tried to address all your concerns in the revised manuscript page 18: line 324 – 325).

Response: Thank you for this suggestion. We have used PACE tool to increase the resolution of the figure 

Authors: Finally, we would like to say thank you for reviewing our work and making insightful suggestions and comments that helped to strengthen our manuscript. We revised the manuscript as necessary

---

## [Decision Letter · Decision Letter 1]

7 Feb 2024

PONE-D-23-08664R1Prevalence of Multidrug Resistance Salmonella species isolated from clinical specimens at University of Gondar Comprehensive Specialized Hospital Northwest Ethiopia: A Retrospective studyPLOS ONE

Dear Dr. Amare,

Thank you for submitting your manuscript to PLOS ONE. After careful consideration, we feel that it has merit but does not fully meet PLOS ONE’s publication criteria as it currently stands. Therefore, we invite you to submit a revised version of the manuscript that addresses the points raised during the review process.

Please review the comments carefully and make changes accordingly. 

We look forward to receiving your revised manuscript.

Kind regards,

Furqan Kabir

Academic Editor

PLOS ONE

Reviewers' comments:

Reviewer's Responses to Questions

**Comments to the Author**

1. If the authors have adequately addressed your comments raised in a previous round of review and you feel that this manuscript is now acceptable for publication, you may indicate that here to bypass the “Comments to the Author” section, enter your conflict of interest statement in the “Confidential to Editor” section, and submit your "Accept" recommendation.

Reviewer #1: (No Response)

Reviewer #2: (No Response)

2. Is the manuscript technically sound, and do the data support the conclusions?

Reviewer #1: Partly

Reviewer #2: Partly

3. Has the statistical analysis been performed appropriately and rigorously? 

Reviewer #1: Yes

Reviewer #2: Yes

4. Have the authors made all data underlying the findings in their manuscript fully available?

Reviewer #1: Yes

Reviewer #2: Yes

5. Is the manuscript presented in an intelligible fashion and written in standard English?

Reviewer #1: Yes

Reviewer #2: Yes

6. Review Comments to the Author

Reviewer #1: While the authors have attempted to elucidate the biochemical profile of the isolates, the current description remains rather generic, lacking specific details for each identity. It is suggested that the authors incorporate a schematic diagram in the manuscript to enhance clarity and provide a more concrete depiction of the biochemical profile associated with each isolate.

The authors have noted the absence of intermediate patterns in the Kirby Bauer AST method used. It is recommended that they include a table in the manuscript presenting the cut-off concentrations for each isolate or group of isolates, thereby offering a more detailed and informative representation of the study findings.

The authors have presented figures related to the samples from which Salmonella was isolated. However, for enhanced clarity, it is suggested that they include a schematic diagram. Additionally, it would be beneficial if the authors could provide information on the total number of samples sent to the lab and specify the count of samples that tested positive for Salmonella. This would contribute to a more comprehensive understanding of the study's outcomes.

Certain table titles still contain errors, such as "at at." It is recommended that the authors carefully review and revise the tables to ensure accuracy and eliminate any such mistakes in the titles.

Reviewer #2: Please see the attached file.

7. PLOS authors have the option to publish the peer review history of their article (what does this mean?). If published, this will include your full peer review and any attached files.

Reviewer #1: No

Reviewer #2: No

---

## [Author Response · Author response to Decision Letter 1]

14 Mar 2024

Review PLOS One

Response letter for manuscript (PONE-D-23-08664R1) 

To Academic Editors and reviewers:

Dear Academic Editor and reviewers,

Authors: We the authors would like to thank the editorial teams of PLOS ONE for quick review process of our manuscript titled “Prevalence of Multidrug Resistance Salmonella species isolated from clinical specimens at University of Gondar Comprehensive Specialized Hospital Northwest Ethiopia”. We appreciate for spending your precious time and forwarding your valuable comments, which have significantly improved our application. We are also grateful for this positive feedback. Please see below, bold, for a point-by-point response to the reviewers. All page numbers refer to the revised manuscript file with tracked changes. All modifications in the manuscript have been highlighted in red.

We are looking forward to hearing from you in due course,

Sincerely,

Azanaw Amare (Corresponding author)

Responses to reviewers’ comments

Reviewer 1

Comments on the re-submitted manuscript

Thank you for responding to the review comments. The reviewer feels that some of the comments have not been well responded to: Please respond appropriately to the following concerns

Reviewer comment: While the authors have attempted to elucidate the biochemical profile of the isolates, the current description remains rather generic, lacking specific details for each identity. It is suggested that the authors incorporate a schematic diagram in the manuscript to enhance clarity and provide a more concrete depiction of the biochemical profile associated with each isolate.

Response: thank you very much for your suggestions and direction. We have tried to illustrate schematic diagram of the biochemical profile of the isolates in figure 1. 

Reviewer comment: The authors have noted the absence of intermediate patterns in the Kirby Bauer AST method used. It is recommended that they include a table in the manuscript presenting the cut-off concentrations for each isolate or group of isolates, thereby offering a more detailed and informative representation of the study findings.

Author response: thank you very much for your remainder. We accept the comment. The isolates with intermediate results detected were very few in numbers, however, for analysis counted as resistant.

Reviewer comment: The authors have presented figures related to the samples from which Salmonella was isolated. However, for enhanced clarity, it is suggested that they include a schematic diagram. Additionally, it would be beneficial if the authors could provide information on the total number of samples sent to the lab and specify the count of samples that tested positive for Salmonella. This would contribute to a more comprehensive understanding of the study's outcomes.

Author response: thank you very much for your direction. We have included schematic diagram count of samples that tested positive for Salmonella species in figure 2.

Reviewer comment: Certain table titles still contain errors, such as "at at." It is recommended that the authors carefully review and revise the tables to ensure accuracy and eliminate any such mistakes in the titles.

Author response: thank very much for your critical view. We have revised such types typing errors throughout the manuscript. 

Reviewer 2 

General comments

Reviewer comment:1.1: Strong points

➢ Study on drug susceptibility patterns of pathogens such as Salmonella species collected from

clinical samples within six years period of time can be used as an evidence for designing

strategies to combat the problem and that makes this study valuable.

➢ The main finding of the study was the prevalence of Salmonella Species which was 0.16%.: ➢ another important part of the study was the prevalence of drug resistance (As low as 16.7%

and as high as 100%) and an overall MDR of Salmonella species which was 53.7%.

Author response: thank you very much for your positive feedback. 

1.2: Limitations

Reviewer comment: ➢ There were topographical, grammatical and punctuation error all over the entire document and I suggest that the manuscript requires language editorial by an expert.

Author response: We are grateful for the suggestion and thank you. We went throughout the entire manuscript to eliminate grammatical and editing mistakes (The manuscript is edited by Solomon Getawa Reta, PhD candidate; at Ariel University).

Reviewer comment: ➢ The study reports was extremely shallow although the data looks to some extent reasonable

Author response: thank very much for your positive feedback.

2. Detail Comments on each component of the manuscript

2.1: Abstract

Reviewer comment: ➢ The title and the abstract good and more or less convey what has been done but documented unsatisfactorily.

Author response: thank very much for your nice comments. We have revised the abstract 

2.2: Introduction

Reviewer comment: ➢ At the beginning of the introduction the authors documented the general characteristics, occurrence and pathogenesis of Salmonella species in two large paragraphs which is not adequate for such a manuscript rather was a general truth that doesn’t correctly introduce the research title which is mainly about drug resistance and MDR. Moreover, no single

evidence about the drug resistance pattern of these bacterial pathogens in Ethiopia and

the Amhara National Regional state was introduced in this section.

Author response: thank very much for your constructive comments. We have revised the introduction as reviewer suggests. (See in page 4&5, line: 92-110)

Reviewer comment:➢ The authors must use quality English while organizing all the parts of the manuscript. 

Author response: We tried eliminating editorial problems, and punctuation errors thought out the revised manuscript

Reviewer comment: In general, the literature review is not adequate and relevant research works that can demonstrate the magnitude of drug resistant salmonella species are not documented.

Author response: We agree with this and have incorporated your comments on (page 5, line: 101-106).

2.3: Methods

Reviewer comment:➢ At the end of the introduction part of the current manuscript, it was explained about the existence of the limited number of studies, lack of a coordinated epidemiological surveillance system, deficient reporting system, and lack of culture facilities in Ethiopia. It was also documented that understanding the prevalence and antimicrobial sensitivity

pattern of Salmonella species will help to choose the right antimicrobial treatment and to

reduce the spread of infection. However, the need to organize the retrospective data for

publication was not adequately justified.

Author response: thank you very much for direction. We have revised as reviewer suggested. (See in page 5, line 107-110) 

Reviewer comment: ➢ The quality control methods used during data collection are not evidenced and don’t guarantee the quality of the data.

Author response: thanks for your feedback. The quality control methods were evaluated during data collection from weekly and daily and weekly quality control record books.

Reviewer comment: ➢ the objective of this retrospective study was described and well defined ‘and’ the entire laboratory methods used were appropriate.

Author response: thank very much for your positive feedback.

Reviewer comment: ➢ in this study, the SOPs of the hospital laboratory were used but no new and specific protocols and algorithms particularly designed and applied in this study.

Author response: we were documented the SOPs of hospital laboratory designed for Salmonella species identification. 

Reviewer comment: ➢ the study was a retrospective study and data was generated from laboratory registration book.

Author response: thank very much for your comments. We have taken the data from Medical Microbiology Laboratory registration book.

Reviewer comment: ➢ the only Software used in data analysis was SPSS version 26 and of course that does not disqualify the research work.

Author response: we have used SPSS version 2 Software to enter and analysis the frequency and factor.

Reviewer comment: ➢ Ethical approval was obtained from School of Biomedical and Laboratory Sciences but not evidenced with an official ethical clearance letter for which the issue number would have been documented. An evidence of the number, date and year of the issue of Letter of permission from the University of Gondar Comprehensive Specialized Hospital would

have been demonstrated.

Author response: thank you very much for your comments. We have stated the reference number, date and year.

2.4: Result

Reviewer comment: ➢ At the beginning of the result section, Socio-demographic, prevalence and frequency of Salmonella species infection were presented and that is mixed. Why? The usual

practice is demographic and clinical characteristics are separately documented and

then compare that with the dependent variable.

Author response: thank you for your nice suggestions. We have separately presented Socio-demographic, prevalence and frequency of Salmonella species infection.

Reviewer comment: ➢ please re-phrase line number 189 to 200.

Author response: thank very much for your direction. We have re-phrased it accordingly. 

Reviewer comment: ➢ the prevalence of Salmonella species was 0.16% which is very low and what is the research finding and which question was answered by this result?

Author response: thank you for your important question. The findings of this study is answered the prevalence of Salmonella species in different clinical sample.

Reviewer comment: ➢ Table one is extremity disorganized and the type of Salmonella bacteria proportion among patients from OPD, IPD and ICU was not known. In this table only age (Vertical column) was compared with other variable which is meaningless. Which one is the dependent variable for this study? Once the dependent variable identified, it should be used to evaluate the association with the independent variables. Table one is very deficient and it must be reorganized.

Author response: thank you very much for your direction. We have reanalysis again and tried to show the association of dependent variable with independent variables (see in page 10 & 12). 

Reviewer comment: ➢ The distribution of Salmonella species from different clinical samples was documented in table 2. What was the major aim of this part? So what is the scientific knowledge gathered from this table? It was reported by different investigators that Salmonella species can cause diseases in the GIT, UT, septicemia, wound infection and even meningitis.

Author response: thank very much for your nice question. The main aim of this table is to know the distribution of salmonella species in different clinical samples. 

Reviewer comment: ➢ In this study, Data was presented in tables and proportions were described but Data was not analyzed properly. For example, there was no even a single evidence of association of the dependent variable with that of the independent variables.

Author response: thank you for your reminder. We are analyzed the association of independent variable and dependent variables (Seen in page 14) 

Reviewer comment: ➢ Trends of antibiotic resistance in Salmonella isolates: this part was

interesting and the most important part of the manuscript but not analyzed properly. First, the authors must learn how to determine “trend” as a statistical term rather than comparison of the percentages. They can demonstrate the trend of AMR over time using graphs and determine if that is a statistically significant increases/decrease in AMR to the commonly used antibiotics. I advise the authors to discuss with statisticians how to determine trend of the AMR and MDR Salmonelossis over the six years.

Author response: thank very much your valuable comments. We have consulted the statisticians to analyses the trend trends of antibiotic resistance in Salmonella isolates and corrected as reviewer suggestion (see in page 16). 

Reviewer comment: ➢ In general, the results of the current study were not clearly presented and there is either very limited result or not at all that reflect the objectives of the study.

Author response: thank you for your comments. We revised the manuscript to relate with the objective. 

Reviewer comment: ➢ In this study no new knowledge explained the study question except the patterns of drug resistance of the salmonella species.

Author response: thank you very much for comments. We try to show the prevalence of and MDR Salmonella isolates. 

2.5: Discussion

Reviewer comment: ➢ The discussion shall begin with some findings about Salmonellosis Globally, in Africa and Ethiopia then followed by one of the finding of the current study but that was not done here. I strongly advice the authors to read, understand and look how to write discussion.

Author response: Thank you for your input. We tried to address all your concerns in the revised manuscript on (page 17: line, 312-318)

Reviewer comment: ➢ Line number 33 to 35: Salmonella infection versus age of the current study was compared with other studies and that was good. But, the way they are compared was not appropriate and the final reason presented on line number 35 and 36 is very immature. What does this discussion add on knowledge or practice in the pathogenesis, diagnosis, treatment and prevention of the disease? This kind of discussion is totally unacceptable. The authors must generate some reasons for the variation or demonstrate some of the reasons documented by literatures.

Author response: thank you for your constrictive comments. We have revised it as reviewer indicated (see in page 17, line 321-324). 

Reviewer comment: ➢ Line number 37 to 43: This was also presented very poorly. What is the scientific evidence here for comparable, higher or lower prevalence of the Salmonella species at different study areas in Ethiopia and abroad? Otherwise, the discussion is meaningless!!

Author response: thank you very much for your nice view. We revised it as reviewer suggested ( see in page 17 line, 328-331)

Reviewer comment: ➢ Line number 43 to 48: most prevalent Salmonella species was S. enterica subsp. arizonae (51%) followed by S. paratyphi A (22%) and S. typhi 14.6%. This study is concordant with the study in Sagamu, Nigeria, but what was the result documented in Nigeria. Here, the reference was presented but there was no evidence for the similarity with the current study and that is totally unacceptable. Again I am forced to ask the authors to read how to organize a discussion and work again???

Author response: thank you for your critical comments. We have revised and removed it accordingly. 

Reviewer comment: ➢ Line number 53 to 61 looks good but need to be rephrased again!

Author response: Thank you very much for reminder. We have rephrased it. 

Reviewer comment: ➢ Line number 62 to 95: the antimicrobial resistance and MDR part would have been discussed better but this is a simple comparison with no convincing evidence or reason. Nothing was discussed about the factors that are highly valuable in drug resistance in Salmonellosis in general and in each of the species in particular.

Author response: thank very much for comments. We have corrected it as reviewer suggests. 

Reviewer comment: ➢ The limitation of the study may not be appropriate as the focus of the study was different from what was documented as limitations.

Author response: thank you very much for your comments. We have revised the limitations part as the reviewer suggestions. 

2.6: Conclusion and Recommendations

Reviewer comment: ➢ The conclusion was good and it was based on the findings of the study but the recommendation was not based on the findings. For example one of the recommendation was stated as “Medical providers should use caution when administering antibiotics to patients with suspected of Salmonellosis during appropriate treatment” which is always true for any type of medication. That is not Unique to this outcome. Other recommendations are not also appropriate which means they are not study outcome based.

Author response: thank very much for you great full suggestion. We have corrected the conclusion and recommendations as reviewer suggested. 

2.7: References

Reviewer comment: ➢ There are 

---

## [Editor Report · Decision Letter 2]

20 Mar 2024

Prevalence of Multidrug Resistance Salmonella species isolated from clinical specimens at University of Gondar Comprehensive Specialized Hospital Northwest Ethiopia: A Retrospective study

PONE-D-23-08664R2

Dear Dr. Amare,

We’re pleased to inform you that your manuscript has been judged scientifically suitable for publication and will be formally accepted for publication once it meets all outstanding technical requirements.

Kind regards,

Furqan Kabir

Academic Editor

PLOS ONE
---

## [Editor Report · Acceptance letter]

26 Apr 2024

PONE-D-23-08664R2 

PLOS ONE

Dear Dr. Amare, 

I'm pleased to inform you that your manuscript has been deemed suitable for publication in PLOS ONE. Congratulations! Your manuscript is now being handed over to our production team.

Kind regards, 

on behalf of

Dr. Furqan Kabir 

Academic Editor

PLOS ONE